# Efficacy of short-course treatment for prevention of congenital transmission of Chagas disease: A retrospective cohort study

**Guillermo Moscatelli**[1,2]\*, **Samanta Moroni**[1,2], **Juan Carlos Ramírez**[2], **Belén Warszatska**[1], **Lascano Fernanda**[1], **Nicolás González**[1], **Andrés Rabinovich**[2], **Jaime Altcheh**[1,2]

1 Servicio de Parasitología y Chagas, Hospital de Niños "Ricardo Gutiérrez", Buenos Aires, Argentina,
2 Instituto Multidisciplinario de Investigaciones en Patologías Pediatricas (IMIPP)- CONICET-GCBA, Buenos Aires, Argentina

\* gfmoscatelli@yahoo.com.ar

## Abstract

### Background

In regions with controlled vector transmission of *T. cruzi*, congenital transmission is the most frequent route of infection. Treatment with benznidazole (BZ) or nifurtimox (NF) for 60 days in girls and women of childbearing age showed to be effective in preventing mother to child transmission of this disease. Reports on short-course treatment ($\leq$30 days) are scarce.

### Methods

Retrospective cohort study. Offspring of women with Chagas disease who received short-course treatment ($\leq$30 days) with BZ or NF, attended between 2003 and 2022, were evaluated. Parasitemia (microhaematocrit and/or PCR) was performed at <8 months of age, and serology (ELISA and IHA) at $\geq$8 months to rule out congenital infection.

### Results

A total of 27 women receiving $\leq$30 days of treatment and their children were included in this study. NF was prescribed in 17/27 (63%) women, and BZ in 10/27 (37%). The mean duration of treatment was 29.2 days. None of the women experienced serious adverse events during treatment, and no laboratory abnormalities were observed. Forty infants born to these 27 treated women were included. All newborns were full term, with appropriate weight for their gestational age. No perinatal infectious diseases or complications were observed.

### Discussion

Several studies have shown that treatment of infected girls and women of childbearing age for 60 days is an effective practice to prevent transplacental transmission of *T. cruzi*. Our study demonstrated that short-duration treatment ($\leq$30 days) is effective and beneficial in preventing transplacental transmission of Chagas disease.

**Data Availability Statement:** All data are in the manuscript and/or supporting information files.

**Funding:** The author(s) received no specific funding for this work.

**Competing interests:** The authors have declared that no competing interests exist.

## Author summary

Reports on short-term treatment (≤30 days) with benznidazole or nifurtimox for Chagas disease in girls and women of childbearing age are scarce. Numerous studies have previously shown that treatment of infected girls and women of childbearing age for 60 days is an effective strategy to prevent transplacental transmission of *T. cruzi*. However, our study provides convincing evidence that a shorter duration of treatment (≤30 days) is not only effective but also beneficial in preventing transplacental transmission of Chagas disease. Our research demonstrated unequivocally that there was no congenital infection in the offspring of the 27 women who received a 30-day treatment regimen. This was confirmed by direct testing at birth and serological confirmation after 8 months of life.

## Introduction

Chagas disease, caused by the zoonotic parasite *Trypanosoma cruzi* (*T. cruzi*), is endemic to the Americas. In areas where vector transmission is under control, congenital transmission represents the main route of infection [1]. The prevalence of Chagas disease among pregnant women in Latin America varies from 5% to 40%, depending on the geographical location. It is estimated that approximately 2 million women of reproductive age in the Americas are infected with *T. cruzi*, and 3 to 10% of them will transmit the infection to their fetuses through the transplacental route, resulting in an annual birth of 9000 infected infants [2]. Only three countries (Argentina, Chile, and Uruguay) have implemented national policies to screen all pregnant women for *T. cruzi* infection [2].

Previous studies have demonstrated the efficacy in the prevention of transplacental transmission through a 60-day treatment regimen with benznidazole (BZ) or nifurtimox (NF) in girls and women of childbearing age, both during the acute and chronic phases [3–6].

However, currently, the administration of treatment during pregnancy is not recommended due to the lack of comprehensive fetal safety data regarding available drugs [7].

Building upon the proven efficacy of treatment in women of childbearing age for preventing congenital Chagas (CHC), as well as the effectiveness of treatment in infants, the ETMI Plus initiative (Elimination of mother-to-child transmission-plus) [8] proposes the following strategic actions to eliminate mother-to-child transmission: achieving a screening coverage of ≥90% for pregnant women, attaining a screening coverage of ≥90% for newborns born to seropositive mothers, and ensuring a treatment coverage of ≥90% for seropositive mothers. Infected children should receive treatment following confirmation of their infection.

Recent studies have suggested that shorter treatment durations, lasting fewer than 30 days, might also be effective in the treatment of Chagas disease. However, there is limited evidence regarding the effectiveness of short-duration treatments (≤30 days) in women of childbearing age for preventing CHC. Therefore, it is crucial to evaluate whether such abbreviated treatment regimens can effectively prevent transplacental transmission of *T. cruzi*.

## Objective

To describe the efficacy of short (≤ 30 days) treatment with BZ or NF in a cohort of girls and women of childbearing age for the prevention of mother-to-child transmission of Chagas disease.

## Materials and methods

### Ethics statement

The Research Committee and the Ethics Review Committee of "Ricardo Gutiérrez" Children's Hospital, Buenos Aires, Argentina approved the study. The reference number of the study is CEI 16.29. Written informed consent and/or permission from the participant and/or legally authorized parent(s) or representative(s) was obtained prior to selection for this study in accordance with the participant's age and local country regulations.

### Study design

Retrospective cohort study.

### Population

Children of women with Chagas disease who received short-term treatment, attended between 2003 and 2022, at the Parasitology and Chagas Service of "Ricardo Gutiérrez" Children's Hospital in Buenos Aires.

### Inclusion criteria

Children of mothers who received treatment with BZ or NF for less than or equal 30 days. Residing in Buenos Aires city, an area without vector transmission. Not having visited endemic areas after treatment.

The following data were collected from maternal medical records: demography, treatment received, *T. cruzi* serology (before treatment and every six months thereafter for a minimum of 3 years), parasitological studies (real-time polymerase chain reaction—*T. cruzi* qPCR -) before and at the end of treatment, clinical course, and laboratory tests.

The serological techniques used were indirect hemagglutination (IHA) (Chagatest IHA, Wiener Laboratory, Argentina) and lysate ELISA (ELISA, Wiener Laboratory, Argentina). IHA titers $\geq 1/16$ and ELISA R $\geq 1.1$ (R: OD sample/OD cut-off) were considered positive.

### *T. cruzi* qPCR studies

Samples from peripheral blood mixed with an equal volume of 6M guanidine hydrochloride EDTA 0.2M buffer, pH 8.00. DNA was obtained using the High Pure PCR Template Preparation kit (Roche Diagnostics Corp., Indianapolis, IN). A duplex real-time PCR (qPCR) assay targeting *T. cruzi* satellite DNA and human RNase P gene was performed with 5 μL of DNA extract in a final volume of 20 μL, using the FastStart Universal Probe Master Mix (Roche Diagnostics GmbHCorp., Mannheim, Germany) and the TaqMan RNase P Control Reagents Kit (Applied Biosystems, Foster City, CA), as previously described [9].

Children born to these treated women underwent evaluation to rule out congenital infection: parasitemia by direct parasitological method (microhematocrit) and/or PCR in those younger than 8 months, and *T. cruzi* serology in those older than 8 months. Children were considered infected (CHC) when parasitaemia was positive or when serology was positive after 8 months of age.

### Statistical analysis

Continuous variables are presented as means with CI95% or medians and interquartile range, and categorical variables as percentages. The disappearance kinetics of serum antibodies were analyzed using survival analysis. Significance levels for analysis were 0.05. Analyses were

performed with R software v3.0 (R Core Team 2018. R Foundation for Statistical Computing, Vienna, Austria. https://www.R-project.org/).

## Results

Maternal data (Table 1): Out of a cohort of 592 women treated in our service, 51 attended the consultation with their children, of whom 24 (47%) received 60 days of BZ or NF treatment and 27 (53%) received 30 days or less of treatment. The latter were included in the analysis together with their children (Fig 1).

Mean age at treatment onset was 27.2 years (range 6 to 42 years). Place of birth: 16 (59.2%) in Bolivia, 9 (33.3%) in Argentina, and 2 (7.5%) in Paraguay. Before treatment, 21 (77.8%) lived in an endemic area. The route of infection was vectorial in 4 patients (14.8%), congenital in 4 (14.8%), and indeterminate in 19 patients (70.4%).

**Table 1. Maternal treatment, diagnosis and follow up.** Infant's outcomes.

| Subject Mother | Days of treatment | Treatment | Age at treatment (years) | Parasitaemia | | Serology | | | | No. of children assessed | Congenital infection |
| --- | --- | --- | --- | --- | --- | --- | --- | --- | --- | --- | --- |
| | | | | PCR pre treatment | PCR end of treatment | ELISA pre treatment (R) | ELISA at 36 months follow-up (R) | HAI pre treatment | HAI at 36 months follow-up | | |
| 1 | 19 | BZ | 21 | Positive | Negative | 12 | 5.1 | 512 | 256 | 3 | NO |
| 2 | 22 | BZ | 20 | Positive | Negative | 12.4 | 3.9 | 1024 | 64 | 3 | NO |
| 3 | 30 | BZ | 6 | Negative | Negative | 2.37 | Neg | 128 | Neg | 1 | NO |
| 4 | 30 | BZ | 9 | Positive | Negative | 3.5 | Neg | 256 | Neg | 1 | NO |
| 5 | 30 | NF | 14 | Negative | Negative | 7 | 2.7 | 2048 | 128 | 2 | NO |
| 6 | 30 | BZ | 20 | Positive | Negative | 12.5 | 3.8 | 4096 | 128 | 2 | NO |
| 7 | 30 | NF | 20 | Positive | Negative | 10.6 | 3.9 | 128 | 32 | 1 | NO |
| 8 | 30 | NF | 23 | ND | ND | 9.1 | 7.3 | 1024 | 256 | 1 * | NO |
| 9 | 30 | NF | 25 | Negative | Negative | 9 | 4.1 | 2048 | 64 | 1 | NO |
| 10 | 30 | NF | 26 | ND | ND | 3.1 | 4.1 | 1024 | 256 | 1 | NO |
| 11 | 30 | NF | 27 | Positive | Negative | 5.8 | 4.8 | 2048 | 256 | 1 * | NO |
| 12 | 30 | NF | 27 | Positive | Negative | 6.8 | 2.8 | 512 | 128 | 2 * | NO |
| 13 | 30 | BZ | 29 | Positive | Negative | 11.6 | 4 | 1024 | 256 | 2 | NO |
| 14 | 30 | NF | 29 | Negative | Negative | 9.3 | ND | 2048 | ND | 1 * | NO |
| 15 | 30 | BZ | 30 | ND | ND | 11.4 | 4 | 1024 | 64 | 2 | NO |
| 16 | 30 | BZ | 30 | Negative | Negative | 12 | 3.5 | 512 | 64 | 1 | NO |
| 17 | 30 | BZ | 31 | ND | ND | 11.4 | 6.1 | 512 | 256 | 3 | NO |
| 18 | 30 | NF | 32 | Positive | Negative | 10.6 | 6.7 | 512 | 256 | 1 | NO |
| 19 | 30 | NF | 33 | Positive | Negative | 7.1 | 3.9 | 256 | 256 | 1 | NO |
| 20 | 30 | NF | 33 | Negative | Negative | 11.8 | 3.1 | 512 | 256 | 2 | NO |
| 21 | 30 | NF | 33 | Positive | Negative | 10 | 2.7 | 256 | 256 | 1 ** | NO |
| 22 | 30 | BZ | 34 | Positive | Negative | 12.1 | 3.7 | 2048 | 256 | 2 | NO |
| 23 | 30 | NF | 34 | ND | ND | 6 | 7.2 | 512 | 256 | 1 * | NO |
| 24 | 30 | NF | 35 | Negative | Negative | 3.1 | 5.3 | 128 | 64 | 1 | NO |
| 25 | 30 | NF | 39 | Positive | Negative | 6.4 | 3.1 | 2048 | 256 | 1 | NO |
| 26 | 30 | NF | 40 | Negative | Negative | 8.9 | 2.1 | 64 | 16 | 1 ** | NO |
| 27 | 30 | NF | 42 | Negative | Negative | 11.4 | 8.3 | 1024 | 512 | 1 | NO |

BZ: benznidazole, NF: nifurtimox, ND: not done, R: OD sample/OD cut-off.

* Has a child infected with *T.cruzi* via the transplacental route prior to treatment.

** Has two children infected with *T.cruzi* via the transplacental route prior to treatment.

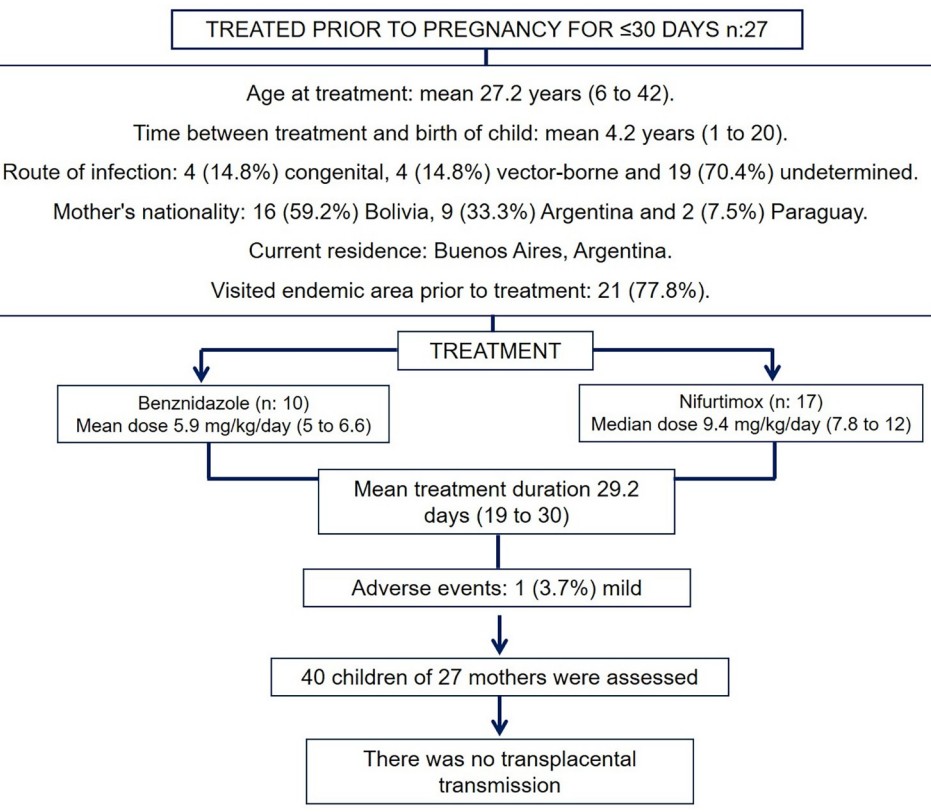

**Fig 1. Sociodemographic, epidemiologic and exposure description of 27 women treated with short treatment with benznidazole or nifurtimox.**

## Treatment

BZ treatment was prescribed in 10/27 patients (37%), with a mean dose of 5.9 mg/kg/day given twice a day (range: 5 to 6.6 mg/kg/day). NF was prescribed in 17/27 women (63%), with a mean dose of 9.4 mg/kg/day three times a day (range: 7.8 to 12 mg/kg/day). The mean duration was 29.2 days (range: 19 to 30 days). None of the subjects experienced serious adverse events or laboratory abnormalities during the course of treatment. Only one patient (3.7%) reported a mild adverse event, presenting with a rash on the body and extremities.

The mean time between the completion of treatment and the birth of the children was 4.2 years (range: 1 to 20 years).

Median follow-up was 5 years (IQR 25–75 4–8). A progressive reduction was observed for ELISA and IHA. After treatment follow-up, showed a 54.5% decrease in antibody titers measured by ELISA and 86% decreased measured by IHA (p<0.001, Wilcoxon test). In 2 patients (7.4%) who received treatment at 6 and 9 years of age, negative seroconversion was observed at 3 years of follow-up (Fig 2).

Pretreatment *T. cruzi* qPCR studies yielded positive results in 13 out of 22 patients (59%). However, after completion of treatment, all patients tested negative, and this negativity persisted throughout the follow-up period (p<0.001, Chi-squared test). In addition, women who initially tested negative for qPCR remained negative during the follow-up assessments.

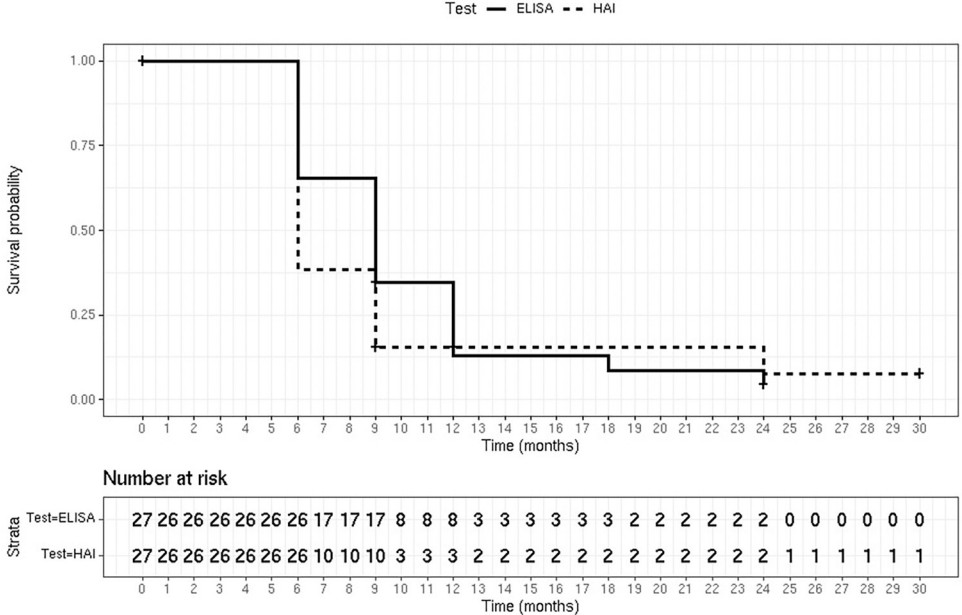

**Fig 2. Survival curve time to negative serology.**

## Infants' outcomes (Table 1)

A total of 40 newborns from 27 mothers were included in the study. Among them, 7 mothers gave birth to 2 children, and 3 mothers gave birth to 3 children. All newborns were full term and had appropriate weight for their gestational age. No perinatal infectious diseases or complications were observed. Direct parasitemia test (microhematocrit test) performed during the early days of life yielded negative in all cases. qPCR tests were performed in 9 out of 40 infants before 8 months of age, and in all cases, the results were negative. All infants underwent serological evaluations (ELISA and IHA) after reaching 8 months of age. Serology results were negative for all cases, ruling out *T. cruzi* infection.

## Discussion

A mother with Chagas disease may transmit the infection to subsequent pregnancies, as transplacental transmission can occur during both the acute and chronic phases of the infection. Increased parasitemia during pregnancy, has been described, with a direct relationship between parasitemia levels and transplacental transmission of *T. cruzi* [10].

BZ and NF have been shown to be effective during the acute and early chronic phase of the disease, particularly using 60-day treatment regimens [11–13]. In addition, several studies showed that 60 days treatment of infected girls and women of childbearing age is an effective practice for preventing transplacental transmission of T. *cruzi* [3,4,6], which is currently the main route of transmission in areas without vector transmission or in areas where vector control is only partially effective [2]. The evidence generated by studies on the effectiveness of etiological treatment in girls and women of childbearing age to avoid congenital transmission, has been key for its inclusion in guidelines for diagnosis and treatment for Chagas from both the PAHO as from endemic countries. Furthermore, treatment has the potential to reduce morbidity and mortality caused by cardiac and digestive disorders in these patients [12]. Currently, drug treatment of Chagas disease during pregnancy is not recommended due to the complete lack of data on the fetal safety of available drugs. However, there are reports of pregnant

women with life-threatening acute infection who have been treated without evidence of serious adverse events in both mother and baby, while fetal infection was prevented [14]. We believe that more detailed studies are needed to reinforce this information.

To improve compliance and reduce the risk of adverse events, studies with shorter treatment durations have been proposed. The BENDITA study showed promising results with 30-day regimens in adult patients in the chronic phase [15]. Several studies in adults are currently underway to confirm these results [16,17]. The first prospective clinical trial in children with Chagas disease (the CHICO-SECURE study), comparing 30- and 60-day NF regimens, has recently been published [18]. It showed similar efficacy results with both regimens, especially in children under 2 years of age.

There is limited evidence on the efficacy of short-term (≤30 days) treatments with BZ and NF in preventing mother-to-child transmission of *T. cruzi*. In a previously published series by our group, there were 3 women treated for less than 30 days [3], who were included in the current study. Another study reported 5 women who were treated for less than 30 days, with similar results [6].

In our study, none of the children born to women who received treatment ≤30 days with BZ or NF acquired the infection. This reinforces the benefits of etiological treatment in girls and women of childbearing age for preventing congenital transmission [3–6], even with short course regimens. The importance of short-duration treatments lies in improved compliance and decreased occurrence of adverse events [19]. One limitation of our study is the number of patients included, however there was no transmission of congenital Chagas disease in any case, coinciding with previous reports that used treatment for 60 days. Further studies are necessary in order to confirm these preliminary findings.

Monitoring treatment response is challenging due to the persistence of anti-*T. cruzi* antibodies for many years after treatment. Assessment of therapeutic response is based on the negativization of parasitemia and negative seroconversion or a sero-reduction of more than 20% of anti-*T. cruzi* antibodies detected by conventional serology. This criterion was used in the CHICO-SECURE study that allowed the rescreening in FDA of the pediatric formulation of nifurtimox, and validated in a study of a large cohort of patients treated by our group [13,18,20]. During the follow-up of treated women, we observed an excellent therapeutic response with a significant decrease in antibody titers, exceeding 20% and even achieving seroconversion in two cases with long-term follow-up.

*T. cruzi* parasitaemia assessed by qPCR has been used to monitor treatment response and/ or failure in different clinical trials [20–22]. All patients who tested positive for qPCR at baseline had negative parasitemia at the end of treatment and remained negative in repeated studies during the follow-up period.

The sustained decline in antibody titers measured by IHA and ELISA, along with the negative qPCR results, suggest that a short-duration treatment (≤30 days) was effective for Chagas disease.

## Conclusion

Previously published data showed the benefits of etiological treatment with 60-day regimens in women of childbearing age to prevent congenital transmission (3,4,5,6). In our study, short-course treatment (≤30 days) has demonstrated efficacy and benefits for the prevention of transplacental transmission of Chagas disease. Addressing the prevention of transplacental transmission is of utmost importance, as Chagas disease continues to be a significant public health concern, particularly in Latin America. With millions of women of reproductive age affected by *T. cruzi*, the potential impact of short-duration treatments in preventing transmission to infants is substantial.

## Author Contributions

**Conceptualization:** Guillermo Moscatelli, Samanta Moroni, Nicolás González, Jaime Altcheh.

**Data curation:** Guillermo Moscatelli, Andrés Rabinovich.

**Formal analysis:** Guillermo Moscatelli, Samanta Moroni, Lascano Fernanda, Jaime Altcheh.

**Investigation:** Guillermo Moscatelli, Samanta Moroni, Juan Carlos Ramírez, Belén Warszatska, Lascano Fernanda, Nicolás González, Jaime Altcheh.

**Methodology:** Guillermo Moscatelli, Andrés Rabinovich, Jaime Altcheh.

**Project administration:** Guillermo Moscatelli, Jaime Altcheh.

**Supervision:** Guillermo Moscatelli, Jaime Altcheh.

**Validation:** Guillermo Moscatelli, Nicolás González, Jaime Altcheh.

**Visualization:** Guillermo Moscatelli.

**Writing – original draft:** Guillermo Moscatelli.

**Writing – review & editing:** Guillermo Moscatelli.

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
