## [Decision Letter · Decision Letter 0]

21 Nov 2023

Dear MD Moscatelli,

Thank you very much for submitting your manuscript "Efficacy of short-course treatment for prevention of congenital transmission of Chagas disease: a retrospective cohort study." for consideration at PLOS Neglected Tropical Diseases. As with all papers reviewed by the journal, your manuscript was reviewed by members of the editorial board and by several independent reviewers. In light of the reviews (below this email), we would like to invite the resubmission of a significantly-revised version that takes into account the reviewers' comments. 

In your revision, and in line with the reviewers comments, please ensure to:

- Provide as much detail as possible re the diagnostic tests / approach used

- Comprehensively discuss the limitations of the study

- Outline next steps to firm up the evidence for recommending / advocating for a shorter treatment regimen

We cannot make any decision about publication until we have seen the revised manuscript and your response to the reviewers' comments. Your revised manuscript is also likely to be sent to reviewers for further evaluation.

Sincerely,

Richard Reithinger

Academic Editor

Charles Jaffe

Section Editor

In your revision, and in line with the reviewers comments, please ensure to:

- Provide as much detail as possible re the diagnostic tests / approach used

- Comprehensively discuss the limitations of the study

- Outline next steps to firm up the evidence for recommending / advocating for a shorter treatment regimen

Reviewer's Responses to Questions

**Key Review Criteria Required for Acceptance?**

**Methods**

-Are the objectives of the study clearly articulated with a clear testable hypothesis stated?

-Is the study design appropriate to address the stated objectives?

-Is the population clearly described and appropriate for the hypothesis being tested?

-Is the sample size sufficient to ensure adequate power to address the hypothesis being tested?

-Were correct statistical analysis used to support conclusions?

-Are there concerns about ethical or regulatory requirements being met?

Reviewer #1: - Objectives were clearly articulated, and the study design was appropriate.

- What ELISA did you use? Lysate or recombinant?

- Related to the Ethics clearance, can you provide the study number or ref?

- Sample size is limited. This is a limitation of the study.

- No statistical analysis was included. For instance, were statistically significant the variations in the ELISA reactivity pre- and post-treatment?

Reviewer #2: The objective of this study is clear and has a logical line and clearly articulated.

The design is consistent with the objective. And the study design is satisfactorily developed. 

Line 110: There is an error in the inclusion criterion: children of mothers treated for at least 30 days with etiological treatment for T cruzi infection. Instead it should say less than or equal to 30 days of treatment, to be consistent with the objective of the study.

Being a cohort study, of an observational nature in a specific period of time, the power of the evidence is not evaluated because there are only 27 cases involved.

**Results**

-Does the analysis presented match the analysis plan?

-Are the results clearly and completely presented?

-Are the figures (Tables, Images) of sufficient quality for clarity?

Reviewer #1: The fact that two of the analyzed participants indeed cured the infection is a very interesting observation emphasizing the efficacy of treatment. However, for the purposes of the study these two mothers shouldn´t have been included as being negative to T. cruzi infection at the time of delivery, meaning there was no risk of transmission.

In terms of the PCR results of the children, would it be possible to have a figure showing these? It should depict age and include the Ct values (mean and SD) of the reactions? Same should be made for the women PCR results if they come from qPCR amplification protocol. 

Regarding the serological results of the children, although a negative general result was obtained, could you please include a figure showing it? If possible, it should include the OD values of positive controls for comparison.

Minor:

- Line 207: "... cases have been ..." change for "... there are ...".

Reviewer #2: The data is clear and is shared correctly, as shown in Table 1. Although I recommend including a table where it can be easier to understand the accumulated values, according to type of treatment, days of treatment, age ranges of having received treatment, and other priorities variables in Table 1, although with accumulated data.

**Conclusions**

-Are the conclusions supported by the data presented?

-Are the limitations of analysis clearly described?

-Do the authors discuss how these data can be helpful to advance our understanding of the topic under study?

-Is public health relevance addressed?

Reviewer #1: The low transmission rate of vertical T. cruzi infection poses a major limitation to the study due to the small cohort size of women who received treatment before delivery. If an average 5% vertical transmission rate is considered only 2 child out of the 40 would have been infected. Please can you comment about this in the Discussion?

Statement in lines 235 and 236: can you further clarify if that >20% seroreduction is a recommendation of WHO-PAHO?

In relation to the two cases that showed cure (subject mothers 1 and 2), if they were negative at the time of considering them for the study, their newborns should not have been included. As stated before, it is a great result to ensure that treatment worked, but their children were then born to non-infected women. 

Regarding the treatment evaluation, optimism is highly appreciated, but a more realistic message in relation to the timely evaluation of treatmen response is missing. Lack of bmks for the early assessment of therapeutic efficacy is a major drawback for the follow-up of subjects who recieved anti-parasitic drugs, and sure having to wait for 36 months to address it is not practical at all.

In line 244 you mention "sustained decline" in the serological outcome of treated women, but there are 3/27 (or 3/25) whose antibody evolution reflected an increase by ELISA, which is considered to have a better sensitivity than IHA. Maybe it would be better to reword that sentence, suggesting that a majority of the women treated experienced a decline in the Ab reactivity by ELISA and IHA when evaluated 36 months after administration. But, according to current guidelines, none was really cured except for the first two girls. A short duration of treatment was effective in preventing vertical transmission but whether the women are cured would require further evidences.

What about shortened administration regimes for the infected newborns. Generally, they much better tolerate benznidazole and nifurtimox than adults, but yet it would be greatly beneficial to have shorter administration times. Can you provide a few lines on this matter?

Reviewer #2: The conclusions are supported by the data presented, however it is perceived that the limitations of the type of study and analysis are not clearly described in the discussion. It is important that the authors can propose that more evidence is necessary in this same approach in order to change and/or adapt the recommendations related to the duration of treatment.

**Editorial and Data Presentation Modifications?**

Reviewer #1: - Please use italics for species names like Trypanosoma cruzi.

Reviewer #2: Line 80, remove stitch after Nifurtimox (NF)

Line 87, translate ETMI-plus: Elimination of mother-to-child transmission (EMTCT)-plus

Line 107 : Change and/or include, in addition to Chagas Disease, T. cruzi infection, because the majority of those treated were not sick, they just had the infection. It is also recommended to make this change throughout the entire text where this type of expression is found.

Line 110: There is an error in the inclusion criteria: children of mothers treated for at least 30 days with etiological treatment for T cruzi infection. Instead it should say less than or equal to 30 days of treatment, to be consistent with the objective of the study.

Line 115: While it is said that the follow-up time with serology is at least 3 years, in relation to parasitological tests (real-time polymerase chain reaction - T. cruzi qPCR) it says before and after without precise time, it would be important to include this information.

Lines 199 -203 : It seems important to include that the evidence generated by studies on the effectiveness of etiological treatment in girls and women of childbearing age to avoid congenital transmission, has been key for its inclusion in regulations/guidelines for diagnosis and treatment for Chagas from both the PAHO as from endemic countries.

**Summary and General Comments**

Reviewer #1: The work presents a retrospective cohort study of T. cruzi-infected women who received halved anti-parasitic treatment and evaluates whether this sufficed to avoid vertical transmission of the infection to their offspring. Results are interesting and this kind of studies is greatly needed by the Chagas community. However, it is limited in size, and a more detailed analysis of the diagnostic results should be provided to better support the treatment response statements. In its current shape it might fit better as short report than as an original article format.

Reviewer #2: This manuscript is interesting and important for the neglected diseases and public health community because it contributes to reviewing the evidence on simplifying care procedures and thus improving and expanding access to antiparasitic treatment for Chagas. Furthermore, the relevance of this manuscript is due to its contribution to a specific population, pointing to a possible health policy that considers the elimination of congenital transmission. I recommend for your better understanding and discussion that the suggestions made in the review be included.

PLOS authors have the option to publish the peer review history of their article (what does this mean?). If published, this will include your full peer review and any attached files.

Reviewer #1: No

Reviewer #2: No
---

## [Editor Report · Decision Letter 1]

2 Jan 2024

Dear MD Moscatelli,

We are pleased to inform you that your manuscript 'Efficacy of short-course treatment for prevention of congenital transmission of Chagas disease: a retrospective cohort study.' has been provisionally accepted for publication in PLOS Neglected Tropical Diseases.

Best regards,

Richard Reithinger

Academic Editor

Charles Jaffe

Section Editor

---

## [Editor Report · Acceptance letter]

18 Jan 2024

Dear MD Moscatelli,

We are delighted to inform you that your manuscript, "Efficacy of short-course treatment for prevention of congenital transmission of Chagas disease: a retrospective cohort study.," has been formally accepted for publication in PLOS Neglected Tropical Diseases.

Best regards,

Shaden Kamhawi

co-Editor-in-Chief

Paul Brindley

co-Editor-in-Chief
